# Rhizobacteria Inoculation Effects on Phytohormone Status of Potato Microclones Cultivated In Vitro under Osmotic Stress

**DOI:** 10.3390/biom10091231

**Published:** 2020-08-24

**Authors:** Tatiana N. Arkhipova, Nina V. Evseeva, Oksana V. Tkachenko, Gennady L. Burygin, Lidiya B. Vysotskaya, Zarina A. Akhtyamova, Guzel R. Kudoyarova

**Affiliations:** 1Ufa Institute of Biology, Ufa Federal Research Centre, Russian Academy of Sciences, Prospekt Oktyabrya, 69, 450054 Ufa, Russia; TNArkhipova@mail.ru (T.N.A.); vysotskaya@anrb.ru (L.B.V.); akhtyamovazarina@gmail.com (Z.A.A.); 2Institute of Biochemistry and Physiology of Plants and Microorganisms, Russian Academy of Sciences, Prospekt Entuziastov, 13, 410049 Saratov, Russia; evseeva_n@ibppm.ru (N.V.E.); burygingl@gmail.com (G.L.B.); 3Vavilov Saratov State Agrarian University, Teatralnaya Square, 1, 410012 Saratov, Russia; oktkachenko@yandex.ru

**Keywords:** *Solanum tuberosum*, in vitro culture, *Azospirillum brasilense* Sp245, *Ochrobactrum cytisi* IPA7.2, auxin, cytokinins, abscisic acid, osmotic stress

## Abstract

Water deficits inhibit plant growth and decrease crop productivity. Remedies are needed to counter this increasingly urgent problem in practical farming. One possible approach is to utilize rhizobacteria known to increase plant resistance to abiotic and other stresses. We therefore studied the effects of inoculating the culture medium of potato microplants grown in vitro with *Azospirillum brasilense* Sp245 or *Ochrobactrum cytisi* IPA7.2. Growth and hormone content of the plants were evaluated under stress-free conditions and under a water deficit imposed with polyethylene glycol (PEG 6000). Inoculation with either bacterium promoted the growth in terms of leaf mass accumulation. The effects were associated with increased concentrations of auxin and cytokinin hormones in the leaves and stems and with suppression of an increase in the leaf abscisic acid that PEG treatment otherwise promoted in the potato microplants. *O. cytisi* IPA7.2 had a greater growth-stimulating effect than *A. brasilense* Sp245 on stressed plants, while *A. brasilense* Sp245 was more effective in unstressed plants. The effects were likely to be the result of changes to the plant’s hormonal balance brought about by the bacteria.

## 1. Introduction

Water deficit inhibits plants’ growth and decreases their productivity. Under natural conditions, plants are frequently exposed to low rainfall and elevated concentrations of osmolytes in the soil solution resulting from increased climate aridity. Remedies are needed to counter this increasingly urgent problem in practical farming. One possible approach is to utilize rhizobacteria known to increase plant resistance to abiotic stresses (drought, salinity, suboptimal temperature, toxic metals) [1,2,3,4]. It has been shown that rhizobacteria support plant growth under stressful conditions by improving their mineral nutrition, water relations, resistance to pathogens and antioxidant functions [5,6,7,8,9]. The capacity of rhizoshere bacteria to produce hormones is considered as an important mechanism promoting plant growth and productivity under both favorable and stress conditions [9,10,11,12,13,14,15,16]. Plant hormone signaling plays an important role in many physiological and developmental processes, including stress response. Bacterial effects on plant hormonal systems are important not only in promoting plant growth directly, but also in other aspects of bacterial action on plants. Thus, hormone-mediated stimulation of root growth can improve mineral nutrition and water relations [17]. Furthermore, hormone-mediated stimulation of plant antioxidant systems helps to protect plants against oxidative stress, which accompanies the action of most of the detrimental environmental factors on plants [18]. However, the effects of bacteria on hormone levels in plants have rarely been studied, while the result of bacterial action is likely to be determined by their ability to influence the hormonal balance of plants [9]. Earlier, we isolated strains of bacteria with an increased ability to synthesize cytokinins [1,15] and auxins [16]. Experiments with these bacterial strains showed a relationship between their ability to produce plant hormones and their effect on the growth of wheat and lettuce plants. Field experiments showed increased productivity of wheat plants treated with hormone-producing bacterial strains [10,19]. However, the yield increments caused by the application of these bacterial preparations were not constant and ranged from 40 to 10% depending on vegetation conditions. Since these conditions cannot be controlled in the field experiments, it is difficult to establish the exact effects of external factors on the effectiveness of bacterial preparations. The use of in vitro culture as a model offers advantages in this regard. This method provides strict control of the plant environment and identifies only target factors in the studied model systems. In particular, in vitro culture enables high air humidity [20]. This approach excludes from consideration growth maintenance brought about by limited water losses and allows for a focus on the direct hormonal effects on plant growth. 

The work was aimed at revealing the effects of bacterial inoculation on plant hormonal balance in potato micro-clones in vitro and the assessment of their importance for maintaining plant growth under stress-free conditions and under water deficit imposed with polyethylene glycol (PEG 6000).

## 2. Materials and Methods 

### 2.1. Plant Material

We used potato (*Solanum tuberosum* L. cv. Nevsky) microplants from the in vitro potato microplant collection of the Agronomy Faculty of Vavilov State Agrarian University (Saratov, Russia), obtained by isolation of apical meristems. Cultivar Nevsky is characterized by high environmental plasticity and is cultivated all over the territory of Russia [21].

### 2.2. Characterization of Bacteria

Two bacterial strains from the IBPPM RAS Collection of Rhizosphere Microorganisms (http://collection.ibppm.ru/) were used as inoculants. *Azospirillum brasilense* Sp245 (IBPPM 219) is a facultative endophyte [22] promoting the growth of a wide range of cultivated plants. The strain is moderately salt-tolerant and bacterial growth stops at 300 mM NaCl [23]. *Ochrobactrum cytisi* IPA7.2 (IBPPM 544, RCAM04481) is a natural potato associate isolated from the rhizosphere of potato cv. Nevsky [24]. *O. cytisi* IPA7.2 promotes potato growth and is promising for association with plants. The strain is halotolerant: bacterial growth stops at 750 mM NaCl [25]. Bacteria were grown in liquid synthetic malate salt medium [26] at 35 °C (optimal growth temperature for both strains) in an ES-20/60 incubator shaker (Biosan, Latvia) at 150 rpm. For plant inoculation, overnight cultures were used (OD being about 0.6 at 660 nm). For analysis of phytohormone production, bacterial cultures were incubated for 36 h (exponential phase of growth) and 102 h (stationary phase). Cells were pelleted by centrifugation at 6000× *g* for 10 min (2-16P Sigma Laborzentrifugen GmbH, Osterode am Harz, Germany) and cultural liquids were analyzed. 

### 2.3. Experimental Design

Potato microplants grown in vitro were divided into microcutting with one leaf and auxillary bud, placed into the tubes with liquid Murashige-Skoog (MS) culture media [27] without hormones and cultivated at 24 °C, 60% air humidity, 60 µmol m^−2^ s^−1^ irradiance and 16 h light period. On the 10th day after cutting potato microclones, bacterial suspension was added into the culture media in 10^6^ CFU ml^−1^ concentration. The colony-forming units (CFU) value was determined through serial dilution plating on the solid synthetic malate-salt medium after 3 days of incubation at 35 °C. Five days after inoculation, the starting culture media was substituted with the culture media (MS) containing PEG 6000 in concentration 25 g L^−1^ corresponding to osmotic potential of −0.3 MPа. Water deficit imposed with polyethylene glycol lasted seven days. Bacteria remained on the root surface after the substitution of culture media [28]. These strains have been previously shown to form a stable association with potato plants, and the stability of microbionts was maintained even during the propagation of microclones [25]. Six variants of cultivation conditions were used for the study of the effects of bacteria and water deficit on plants: non-inoculated plants grown with or without PEG; plants inoculated with either *A. brasilense* Sp245 or *O. cytisi* IPA7.2 strains grown with or without PEG (the variants are specified in the table and figures; 30 microclones were used per each cultivation condition). Assessment of the plant state was performed on the basis of measuring mircoplant morphological parameters (stem length and mass, the mass of roots and all leaves of one plants) of 30 microclones per each experimental variant and hormone content (auxin indolyl-3-acetic acid (IAA), abscisic acid (ABA), cytokinins (zeatin and isopentenyl adenine and their ribosides) in the leaves, stems and roots of plants on the 7th day of stress. 

### 2.4. Determination of Phytohormones

Hormones were extracted from freeze-dried samples of leaves, stems, and roots of 5 plants with 80% ethanol and were quantified using enzyme-linked immunosorbent assay (ELISA) after their solvent partitioning and purification. Ethanol extraction was omitted in the case of bacterial culture media. IAA and ABA were purified as described previously [29]. In short, they were extracted with diethyl ether (at pH 2–3) from the aqueous residue of ethanol extracts, then transferred into 1% sodium hydrocarbonate, and secondary re-extraction was performed with diethyl ether from aqueous extract acidified with HCl (at pH 2–3). The volumes of extractants were reduced at each stage of extraction and re-extraction, enabling the release of the extract from related compounds. IAA and ABA were analyzed using corresponding specific serum as described previously [30]. Cytokinins in aqueous residue were concentrated on C-18 cartridge (Waters Corporation, Milford, MA, USA), separated with the help of thin-layer chromatography as described previously [31]. Zeatin and isopentenyl adenine derivatives (free bases and ribosides) were assayed in the eluates of the zones corresponding to the position of standards with antibodies raised against ribosides of zeatin and isopentenyl adenine as described [30]. The reliability of the method was confirmed by comparison of its results with the data obtained with the help of HPLC combined with mass-spectrometry [31,32].

### 2.5. Statistics

Data were processed by means of one-way analysis of variance (ANOVA) with Duncan’s multiple range test used to discriminate means (*p* ≤ 0.05) in the case of analysis of morphological parameters (AGROS program package for statistical and biometrical–genetic analysis in plant breeding and selection; Version 2.09; Department of Statistical Analysis, Russian Academy of Agricultural Sciences) and least significant difference (LSD_0.05_) test used for analysis of hormone content with Excel software (Microsoft Corporation, Albuquerque, NM, USA). 

## 3. Results 

### 3.1. Morphological Parameters

Under stress-free conditions (without PEG), *A. brasilense* Sp245 increased leaf mass compared to the control (microclones untreated with bacteria), while the leaf mass value of microclones incoculated with *O. cytisi* IPA7.2 was intermediate between that of the control plants and plants inoculated with *A. brasilense* Sp245 (Table 1). Alongside with increasing leaf mass, *A. brasilense* Sp245 accelerated accumulation of the stem mass. In the presence of bacteria, root mass was not different from that of the control, but was significantly lower in plants treated with *O. cytisi* IPA7.2 compared to *A. brasilense* Sp245. Thus, in the absence of stress, inoculation with *A. brasilense* strain Sp245 was more effective in promoting biomass accumulation by potato microclones than inoculation with *O. cytisi* IPA7.2 (Table 1).

PEG treatment strongly inhibited the growth of microplants, while both strains increased stem elongation and accumulation of leaf biomass in the stressed microplants. Nevertheless, the effect of the *O. cytisi* IPA7.2 strain was better pronounced: the presence of these bacteria led to significantly greater accumulation of leaf biomass when compared to the effect of *A. brasilense* Sp245. Furthermore, in the case of plants growing in solution with PEG, inoculation with *A. brasilense* Sp245 did not influence root mass compared to the plants non-inoculated with bacteria, while *O. cytisi* IPA7.2 increased root biomass accumulation. Thus, the treatment with *O. cytisi* IPA7.2 led to greater growth promotion than *A. brasilense* Sp245 under stress conditions (Table 1).

### 3.2. Hormones in Bacterial Culture Media

During their life activity in vitro, bacteria excreted hormones into the liquid synthetic malate-salt medium. For *A. brasilense* Sp245 and *O. cytisi* IPA7.2 cultures, 36 and 102 h of cultivation corresponded to the early and late stationary phases of growth. For *A. brasilense* Sp245 and *O. cytisi* IPA7.2 cultures, optical density values were 1.26 and 1.52 after 36 h and 1.29 and 1.55—after 102 h, respectively. Of the phytohormones we studied (IAA, ABA, cytokinins), the concentration in the culture media was highest in the case of IAA (Table 2).

IAA concentration increased during cultivation of both strains. *A. brasilense* Sp245 cells were the most active in synthesizing IAA, the content of this hormone increasing about 20 times from 36 to 102 h of cultivation, although the optical density of the bacterial culture did not increase with time and was lower than in the case of *O. cytisi* IPA7.2. The content of cytokinins and ABA in the culture media was much lower (1–2 ng ml^−1^) and did not increase in the process of cultivation.

### 3.3. Effect of Bacteria on the Content of Hormones in the Plants

#### 3.3.1. Auxin

Figure 1 shows that, without PEG, both strains increased IAA content in the leaves and did not influence that in the roots. In the stem, the content of this hormone was decreased by *O. cytisi* IPA7.2., Bacterial treatment resulted in the accumulation of IAA in the stems of stressed microclones (grown with PEG), with *A. brasilense* Sp245 inducing a greater increase in the level of this hormone than *O. cytisi* IPA7.2. Bacterial treatment did not change IAA content in the leaves, and *A. brasilense* Sp245 decreased IAA in the roots of microclones grown in the solution with PEG. 

#### 3.3.2. ABA

Accumulation of ABA (about 3-fold increase in its level) in the leaves of non-inoculated plants grown with PEG was the most noticeable result of the quantitative analysis of this hormone (Figure 2). Without PEG, bacteria caused about a 1.5-fold increase in the level of ABA in the leaves. On the contrary, in the plants grown with PEG, bacterial inoculation resulted in a suppression of the increase in the leaf abscisic acid that was otherwise promoted by PEG in the potato microplants. In the stems of plants subjected to inoculation and/PEG treatment, ABA content was significantly lower than in the non-inoculated plants grown without PEG, except for the variant with PEG treatment and inoculation with *A. brasilense* Sp245. The latter variant of the experiment was also the only one in which a significant change (decline) in ABA content was detected in the roots. At the same time, neither inoculation with bacteria nor osmotic stress acting by itself affected the concentration of ABA in the roots of plants.

#### 3.3.3. Cytokinins

Bacterial treatment increased the content of zeatin derivatives (free bases and ribosides) in the stems of plants grown under stress-free conditions and under the water deficit imposed with polyethylene glycol (Figure 3), the greatest increment being detected in the plants inoculated with *A. brasilense* Sp245. In the leaves of plants grown with PEG, the content of this form of cytokinins was increased to the greatest extent by *O. cytisi* IPA7.2. Without PEG, the highest level of zeatin derivatives was detected in the roots of plants inoculated with *O. cytisi* IPA7.2, and in the variant with PEG, in plants inoculated with *A. brasilense* Sp245.

In the plants that grew under stress-free conditions (without PEG), the accumulation of isopentenyladenine derivatives was the most noticeable effect in the leaves of plants inoculated with *A. brasilense* Sp245, as well as in the stems of plants inoculated with both bacterial strains (Figure 4). The level of isopentenyladenine in the leaves was increased by the treatment with *A. brasilense* Sp245 to less extent in plants grown with PEG than without it.

## 4. Discussion

Previous experiments showed that *Azospirillum brasilense* Sp245 and *Ochrobactrum cytisi* IPA7.2 promoted the growth and adaptive capacity of inoculated potato plants both under in vitro and ex vitro conditions (the planting of inoculated plants in the pots with soil in the Greenhouse as well as in the field experiments). In particular, inoculation with A. brasilense Sp245 resulted in a 1.5-fold increase in plants’ survival and a 45% increase in the productivity of potato plants grown under field conditions [26]. Results of the present experiments suggest that these observations may be due to the bacterial effects on the hormonal system *in planta*. The obtained results show that the presence of bacteria influenced hormonal levels in the plants. The increase in IAA detected in plants inoculated with bacteria may be most easily explained by the ability of bacteria to synthesize this hormone, as well as by its uptake by the plants. This explanation is consistent with the fact that the auxin level in the plants was higher in the presence of *A. brasilense* Sp245 (Figure 1) accumulating more auxins in the culture media (Table 2). None of the tested bacteria influenced IAA content in the roots (Figure 1), which is in accordance with our previous data showing that prolonged cultivation with bacteria increased the concentration of hormones in shoots and not roots [33]. 

Cytokinins and abscisic acid were present in much lower concentrations than IAA in the bacterial culture media. This does not allow us to associate the accumulation of either cytokinins or ABA in inoculated plants with the capacity of bacteria to produce hormones. However, the production of hormones by bacteria is known to increase under the influence of plant components of the culture media [34], as well as under the influence of PEG [35]. Thus, it cannot be ruled out that the bacterial production of ABA and cytokinins may increase with the joint cultivation of bacteria and plants. An alternative explanation of the changes in the content of hormones in plants may be found in the reports showing the effect of bacteria on the capacity of plants themselves to metabolize hormones [2,36,37,38].

Acceleration of stem elongation under the effect of bacteria, found by us in the presence of PEG, is consistent with the increased IAA content in plant stems (Figure 1). These results can be explained by the reports demonstrating the ability of auxins to stimulate stem cell elongation [39]. Auxins are reported to be crucial for abiotic stress signalling [40]. Our data are in accordance with those published previously showing that auxin producing *Bacillus licheniformis* reduced the extent of growth inhibition by drought stress in pepper [41]. The increase in drought resistance by auxins have been also shown in transgenic potato plants overexpressing the Arabidopsis *YUCCA* gene responsible for auxin synthesis [42]. Nevertheless, in the presence of PEG, significantly more auxins accumulated in the stems of the plants treated with *A. brasilense* Sp245 than with *O. cytisi* IPA7.2, while the promotion of stem growth was similar in both of them (Table 1). The similar promotion of stem elongation in the presence of *A. brasilense* Sp245 and *O. cytisi* IPA7.2 despite higher auxin content in the stems of the former may be due to an increased level of ABA in the stems of the plants inoculated with *A. brasilense* Sp245 (Figure 2) (ABA is known to act as an antagonist of auxins in the control of cell elongation) [43].

ABA is considered as one the most vital growth regulators involved in osmotic stress signalling and tolerance [40]. ABA accumulates to high levels during drought stress [17]. The production of this hormone by *Azospirillum lipoferum* increased ABA concentration in inoculated maize (Z. mays) seedlings, resulting in stomatal closure [3]. However, stomatal closure inhibits photosynthesis, leading to the inhibition of plant growth. Consequently, accumulation of ABA in the leaves of osmotically stressed plants observed in the absence of bacteria (Figure 2) could be the cause of inhibition of leaf growth, while suppression of the PEG-induced increase in the leaf abscisic acid by bacterial inoculation may contribute to their faster growth. Our data are in accordance with those showing that *Bacillus subtilis* maintained photosynthesis and growth of Arabidopsis plants by reducing ABA concentration *in planta* [44].

The known ability of cytokinins to maintain plant growth either under normal or stress conditions [45] is important for controlling plant productivity. Cytokinin signalling cascades participate in the transduction of signals that are triggered by osmotic conditions [46]. Bacterial inoculation mainly increased the level of isopentenyl adenine derivatives in the leaves and zeatin derivative in the stems. In the absence of osmotic stress, a relation was detected between a high level of isopentenyl adenine derivatives and leaf mass accumulation in plants inoculated with *A. brasilense* Sp245. In the stem, the highest level of cytokinins was also detected in the case of *A. brasilense* Sp245, although, not in the form of isopentenyl adenine derivatives, but of zeatin. In this case (without PEG), it is also easy to notice the relationship between the high level of zeatin derivatives in the stem of microplants inoculated with *A. brasilense* Sp245 and accumulation of the stem mass. In osmotically stressed plants, a linkage was also detected between the effect of bacteria on the level of cytokinins and activation of leaf growth, although not in the case of *A. brasilense* Sp245, but of *O. cytisi* IPA7.2. Thus, the greatest content of zeatin derivatives was revealed in the leaves of stressed microplants inoculated with *O. cytisi* IPA7.2, which corresponded with the highest level of their leaf mass accumulation. 

Our recent publication reporting on the effects of the same bacterial strains on potato microclones showed a sharp decrease in the leaf content of Chl a in osmotically stressed plants, while *O. cytisi* IPA7.2 helped to preserve the constitutive level of green pigments in the leaves [28]. Maintenance of chlorophyll content in plants treated with *O. cytisi* IPA7.2 may be explained by higher zeatin derivatives in the leaves of these plants, since cytokinins are effective in delaying the breakdown of chlorophyll in stressed plants [47]. It is of interest that under normal conditions (without PEG) it was inoculation with *A. brasilense* Sp245 that contributed to higher content of both chlorophyll a and b [26] and isopentenyl adenine derivatives (Figure 4) in the leaves compared to inoculation with *O. cytisi IPA7.2*. This may be due to the effect of cytokinins on chlorophyll content, which was detected under normal conditions of growing plants in vitro [48]. Thus, both the increase in cytokinins by *A. brasilense* Sp245 under normal conditions, as well as the increased cytokinins in the stressed plants inoculated with *O. cytisi IPA7.2,* are likely to be important for maintaining higher levels of chlorophyll, bringing about the optimization of photosynthesis and plants growth. The detected difference between the strains suggests that *A. brasilense* Sp245 was more efficient under normal conditions, while *O. cytisi* IPA7.2—in stressed plants.

It is also of interest that the accumulation of cytokinins in roots of plants treated with *O. cytisi* IPA7.2 under the stress-free conditions was accompanied by a decreased accumulation of root mass. The detected relationship between the accumulation of cytokinins and the inhibition of root growth is in accordance with the data showing the ability of cytokinins to inhibit the growth and biomass accumulation of the roots [49,50]. Inhibition of root growth under the influence of *O. cytisi* IPA7.2 is likely to decrease root capacity for water and ions uptake that could indirectly reduce the potentially positive effects of these bacteria on the shoot growth in the absence of PEG. It is of interest that, in the presence of PEG, cytokinins did not accumulate in the roots of plants inoculated with *O. cytisi* IPA7.2, which was accompanied by stimulation of the root growth. In this case, cytokinins were likely allocated into the shoots. Unlike the effects detected under stress-free conditions, cytokinins accumulated in the roots of stressed plants inoculated with *A. brasilense* Sp245, which corresponded with the absence of a stimulating effect on the root growth and lower activation of the plant growth in general. At present, the mechanisms regulating cytokinins transport from roots to shoots are actively studied [50,51]. Membrane transport of cytokinins has been shown to be important for cytokinins outflow from root to shoots [31]. However, the mechanisms regulating these processes are still insufficiently understood. The lack of clear understanding of the mechanisms of cytokinin transport from roots to shoot does not allow us to explain how O. cytisi IPA7.2 caused the accumulation of cytokinins in roots under normal conditions and their supposed export to shoots in PEG-treated plants. Nevertheless, this phenomenon is interesting in itself, and the effect of bacteria on the distribution of cytokinins between shoots and roots requires further study. At any case, the absence of cytokinin accumulation in the roots of PEG-treated plants inoculated with *O. cytisi* IPA7.2 was seemingly important to provide a greater stimulating effect on plant growth.

## 5. Conclusions

Thus, we have shown that the stimulating effect of bacteria on the growth and adaptation of plants is associated with their effects on the hormonal content in plants. Bacterial inoculation resulted in increased concentrations of auxin and cytokinin in the leaves and stems and in a suppression of the PEG-induced increase in the leaf ABA. These effects were obviously important for plants’ growth promotion by bacteria under both normal or stress conditions. The clearer effect of *A. brasilense* Sp245 in the absence of stress and of *O. cytisi* IPA7.2 in the presence of PEG was obviously a consequence of the special effects of bacteria on the hormonal system under these conditions, which may be due to the difference in the ability of the bacteria of these two species to function in stressful and normal environments. Further experiments are needed to clarify the mechanisms of bacterial effects on hormonal transport from roots to shoots depending on conditions of their growth. The data reported in the present and previous articles [28] show that *A. brasilense* Sp245 is sufficiently promising for use as a constituent of biopreparation under a moderate climate, while *O. cytisi* IPA7.2 may be successfully used under salinity and drought conditions. A study of the bacterial effects on plant hormonal systems may be useful for the optimization of their application under different environment.

## Figures and Tables

**Figure 1 biomolecules-10-01231-f001:**
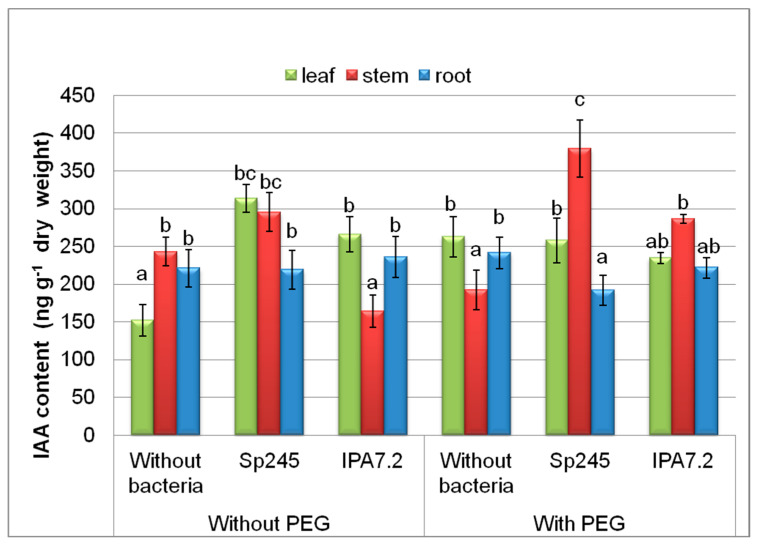
The effect of PEG and microplants inoculation with bacterial strains (*A. brasilense* Sp245 and *O. cytisi* IPA7.2) on the content of IAA in organs of potato plants cultivated in vitro. Means ± SE (*n* = 6) marked with similar Latin letters do not differ significantly according to the LSD-test (*p* ≤ 0.05).

**Figure 2 biomolecules-10-01231-f002:**
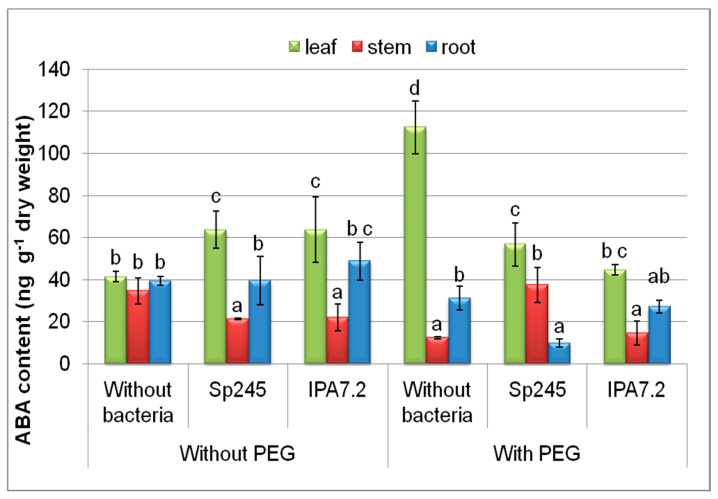
The effect of PEG and microplants’ inoculation with bacterial strains (*A. brasilense* Sp245 and *O. cytisi* IPA7.2) on the content of ABA in organs of potato plants cultivated in vitro. Means±SE (*n* = 6) marked with similar Latin letters do not differ significantly according to the LSD-test (*p* ≤ 0.05).

**Figure 3 biomolecules-10-01231-f003:**
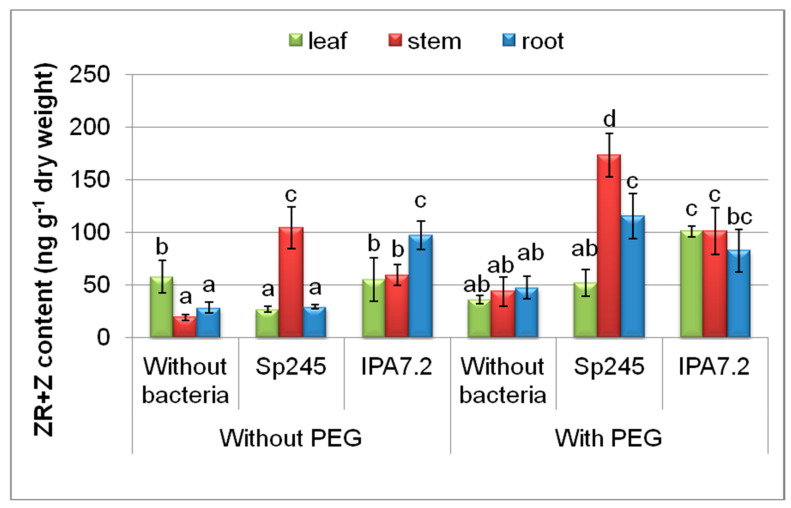
The effect of PEG and microplants’ inoculation with bacterial strains (*A. brasilense* Sp245 and *O. cytisi* IPA7.2) on the content of zeatin derivatives (ZR+Z) in organs of potato plants cultivated in vitro. Means ± SE (*n* = 6) marked with similar Latin letters do not differ significantly according to the LSD-test (*p* ≤ 0.05).

**Figure 4 biomolecules-10-01231-f004:**
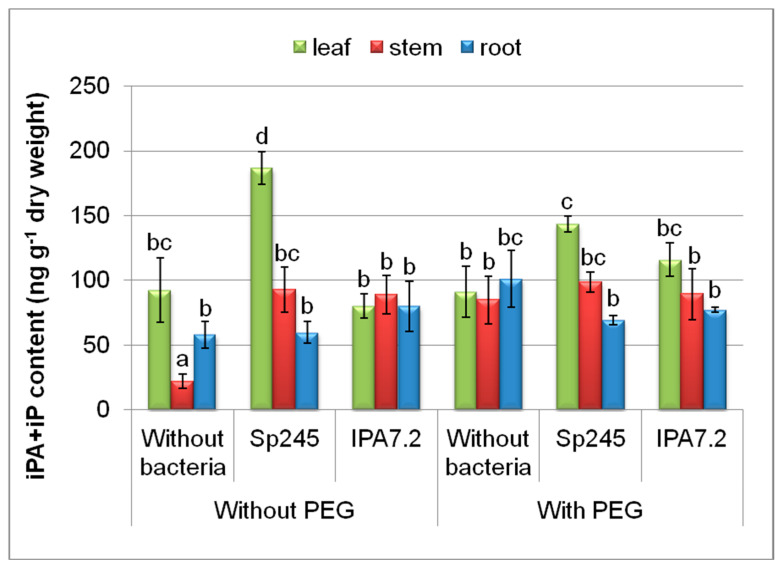
The effect of PEG and microplants’ inoculation with bacterial strains (*A. brasilense* Sp245 and *O. cytisi* IPA7.2) on the content of isopentenyl derivatives (isopentenyl adenine (iP) and isopentenyl adenosine (iPA)) in organs of potato plants cultivated in vitro. Means ± SE (*n* = 6) marked with similar Latin letters do not differ significantly according to the LSD-test (*p* ≤ 0.05).

**Table 1 biomolecules-10-01231-t001:** The effect of bacteria (*A. brasilense* Sp245 and *O. cytisi* IPA7.2) morphological parameters of plants grown with or without PEG measured after 22 days of microplant cultivation.

Variants		Stem Length mm	Fresh Mass, mg		
			Stem	Leaves	Roots
Without PEG	Without bacteria	59.7 ^bc*^	119.0 ^b^	115.0 ^c^	97.0 ^cd^
	+Sp245	65.4 ^c^	153.0 ^c^	152.0 ^d^	112.0 ^d^
	+IPA7.2	57.9 ^b^	121.0 ^b^	141.0 ^cd^	77.0 ^bc^
With PEG	Without bacteria	46.1 ^a^	62.1 ^a^	43.7 ^a^	37.7 ^a^
	+Sp245	59.6 ^bc^	56.6 ^a^	85.3 ^b^	33.0 ^a^
	+IPA7.2	64.0 ^bc^	56.9 ^a^	119.0 ^c^	87.0 ^bc^

* Variants marked with similar Latin letters do not differ significantly according to Duncan’s multiple range test (*n* = 30, *p* ≤ 0.05).

**Table 2 biomolecules-10-01231-t002:** Concentration of indolyl-3-acetic acid (ng ml^-1^) in the nutrient liquid synthetic malate-salt medium after cultivation of the bacteria.

Bacterial Strains	Duration of Cultivation	
	36 h	102 h
*O. cytisi* IPA7.2	6.3 ^a*^	12.1 ^b^
*A. brasilense* Sp245	9.7 ^a^	192.4 ^c^

*Variants marked with similar Latin letters do not differ significantly according to Duncan’s multiple range test (*n* = 6, *p* ≤ 0.05).

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
