# Peer review of "Rhizobacteria Inoculation Effects on Phytohormone Status of Potato Microclones Cultivated In Vitro under Osmotic Stress"

_biomolecules, 2020, doi:10.3390/biom10091231_

Round 1

Reviewer 1 Report

The authors report interesting results about the investigation of the Azospirillum brasilense Sp245 and Ochrobactrum cytisi IPA7.2 inoculation effects on in vitro potato microplants. Effects were evaluated by plant growth parameters and hormones contents under normal and under water deficit stress induced through polyethylene glycol (PEG 6000). The manuscript presentation is clear. The aim of the study and materials and methods have been well presented. However, experimental design, data collection and statistical analysis details presentation should be ameliorated.

Leaving aside the statistical part to be clarified, results have been reported clearly with a good discussion section. However, discussion and conclusion sections should be ameliorated. What are the implications of the findings reported? Are there some limitations? Future research perspectives should be mentioned.

Also, I suggest to revise the title as follows: “Rhizobacteria inoculation effects on phytohormone status of potato microclones cultivated in vitro under osmotic stress”

I suggested other changes in the pdf file attached.

Author Response

Our most sincere thanks to the reviewer for kind words about our manuscript, its attentive and detailed comments! We highly appreciate such great efforts and tried to do our best to follow all the recommendations.

  1. Reviewer’s remark: It was recommended to write in the discussion about the implications of the findings reported and future research perspectives.

Response: In accordance with this we added sentences to the previous variant of Conclusion and it now sounds as: “Further experiments are needed to clarify mechanisms of bacterial effects on hormonal transport from roots to shoots depending on conditions of their growth. The data reported in the present and previous articles [26] show that. A. brasilense Sp245 is sufficiently promising for the use as a constituent of biopreparation under moderate climate, while O. cytisi IPA7.2 may be successfully used under salinity and drought. Study of the bacterial effects on plant hormonal system may be useful for optimization of their application under different environment.” Furthermore, several new sentences were also added to the Introduction: “Plant hormone signaling plays an important role in many physiological and developmental processes including stress response. Bacterial effects on plant hormonal system are important not only in promoting plant growth directly, but also in other aspects of bacterial action on plants. Thus hormone-mediated stimulation of root growth can improve mineral nutrition and water relations [18] Furthermore, hormone-mediated stimulation of plant antioxidant systems, helps to protect plants against oxidative stress, which accompanies action of most of detrimental environmental factors on plants [19].

We also made some more changes in the Discussion section, which may be tracked.

  1. Reviewer’s remark: It was suggested to revise the title as follows: “Rhizobacteria inoculation effects on phytohormone status of potato microclones cultivated in vitro under osmotic stress”

Response: the suggested variant was accepted.

The rest of recommendations were extracted from the attached file. Thus

  1. Lines 85-86 in the revised version; lines 74-75 in the original variant. «For plant inoculation, overnight cultures were used.» Reviewer wanted to know “Which was bacterial density obtained?»

Response: We added in the brackets at the end of the sentence “(OD being about 0.7 at 660 nm)”

  1. Line 95-96. It was advised to specify how CFU was measured.

In accordance we added that “Colony forming units (CFU) value was determined through serial dilution plating on the solid synthetic malate-salt medium after 3 days of incubation at 35°C. “

  1. Lines 106-107. The question was «How many clones per each cultivation conditions?»

-We added that 30 microclones were used per each cultivation conditions

  1. Lines 94-97. The question was «How many replications per each experiment?».

 -We added that morphological characteristics were measured in 30 microclones per each experimental variant. Number of replicates is also added to figure legends.

  1. Lines 115- 116. Reviewers’ remark: “Authors should revise this section thoroughly, describing which data have been processed by Duncan's multiple range test and which by LSD-test. Also, add details of statistical software used.”

In accordance with this remark we revised statistics paragraph: “Data were processed by means of one way analysis of variance (ANOVA) with Duncan’s multiple range test used to discriminate means (p≤0.05) in the case of analysis of morphological parameters (AGROS program package for statistical and biometrical–genetic analysis in plant breeding and selection) and least significant difference (LSD0.05) test used for analysis of hormone content with Excel software (Microsoft Corporation).” It is also specified, which test was used in the table and figure legends

  1. Line 159-163. Reviewer’s question «Does authors have data about bacterial density (CFU mL -1) of both strains at 36 h and 102 h? This results could add information about bacterial density necessary for each strain to provide the reported concentration of hormones.»

We added that “For A. brasilense Sp245 and O. cytisi IPA7.2 cultures, optical density values were 1.26 and 1.52 after 36 hours of cultivation and 1.29 and 1.55 - after 102 hours, respectively.” And this really allowed as (as reviewer expected ) to add to the sentence (lines 171-172) telling that A. brasilense Sp245 cells were the most active in synthesizing IAA that this was “although optical density of the bacterial culture did not increase with time and was lower than in the case of O. cytisi IPA7.2”

  1. Reviewer suggested including errors into the figures and this was done (see figures 1-4).
  2. The last recommendation was to shorten several sentences in the Discussion. This was done by dividing them into two sentences. It remains to hope they sound clearer now.
  • Lines 238-232. However, the mechanisms regulating these processes are still insufficiently understood. The lack of clear understanding of the mechanisms of cytokinin transport from roots to shoot does not allow us to explain how O. cytisi IPA7.2 caused the accumulation of cytokinins in roots under normal conditions and their supposed export to shoots in PEG-treated plants.
  • Lines 339-341. Bacterial inoculation resulted in increased concentrations of auxin and cytokinin in the leaves and stems and in a suppression of the PEG-induced increase in the leaf ABA. These effects were obviously important for plants growth promotion by bacteria under both normal or stress conditions.
  1. All other remarks concerning lettering (capital letters and Italics) and deleting repeated words were gratefully accepted, and changes made in accordance (sorry for having been not attentive enough).

Reviewer 2 Report

The paper brings some new information to the knowledge about the phytohormone status of potato microclones under osmotic stress in vitro. The experiments were carefully planed and well performed, however, there are a few points to be added or explained. Attention should be paid also to the language. In total, I found that the study advance the subject area.

I suggest improvements in the light of following comments:

Very little attention was paid to the osmotic stress signalling and its crosstalk with phytohormone-mediated signalling pathways in plants.

Also, this study just reports on a single experiment (in vitro) but it would be interesting to see its correlation with Greenhouse study. 

In Introduction section “line 37” provide specify the name of stresses 

Its not completely true. There is different study about mechanism of stress resistance "line 37-38" and could be refined by adding important information.

Make a consistency throughout the manuscript by using single notation for each word for e.g. stress-free condition/without PEG/in the absence of PEG was used at different places "119, 192; table 1". Same for PEG TREATED/mild osmotic stress "line 170" as well as IAA content/auxin content

On what basis have you shortlisted the time points for the in-vitro extraction of bacterial hormones, Table 2?

Its not clear, Reconstruct the line " Furthermore, unlike A. brasilense........." Line 136-138.

Authors need to shed more light in discussion section on role of specific phytohormone as far as possible in the light of already reported information/mechanism of each stress responsive phytohormone in various plants.

Author Response

We are most grateful to the reviewer for attentive reading of our article and valuable comments. We tried to do our best to follow them.

  1. “Very little attention was paid to the osmotic stress signalling and its crosstalk with phytohormone-mediated signalling pathways in plants.”

– in accordance with this remark we added 3 sentences into discussion concerning auxins, cytokinins and ABA:

  • Lines 258-259.”Auxins are reported to be crucial for osmotic stress signalling [40]”
  • Lines 270-271. “ABA is considered as one the most vital growth regulators involved in osmotic stress signalling and tolerance [40].”
  • Lines 283-284. “Cytokinin signalling cascades participate in transduction of signals that are triggered by osmotic conditions [45, and references therein]”.

More information is also added according to the last remark (see below)

  1. “Also, this study just reports on a single experiment (in vitro) but it would be interesting to see its correlation with Greenhouse study”.

 – We are sorry that our original variant of the manuscript left the impression that effects of bacteria were studied only in vitro. To rectify this we added to the Discussion (lines 234-240) that “Previous experiments showed that Azospirillum brasilense Sp245 and Ochrobactrum cytisi IPA7.2 promoted growth and adaptive capacity of inoculated potato plants both under in vitro and ex vitro conditions (planting of inoculated plants in the pots with soil in the Greenhouse as well as in the field experiments). In particular, inoculation with A. brasilense Sp245 resulted in 1.5 fold increase in plants survival and 45 % increase in productivity of potato plants grown under field conditions (Tkachenko et al., 2015).”

  1. “In Introduction section “line 37” provide specify the name of stresses”

– In accordance with this remarks we added specification of the stresses in the brackets (lines 39-41): “One possible approach is to utilize rhizobacteria known to increase plant resistance to abiotic stresses (drought, salinity, suboptimal temperature, toxic metals)”

  1. “Its not completely true. There is different study about mechanism of stress resistance "line 37-38" and could be refined by adding important information

 – In accordance with this remark we added following sentences: “It has been shown that rhizibacteria support plant growth under stressful conditions by improving their mineral nutrition, water relations, resistance to pathogens and antioxidant functions [5-9]. The capacity of rhizoshere bacteria to produce hormones is considered as an important mechanism promoting plant growth, and productivity under both favorable and stress conditions [9-17]. Plant hormone signaling plays an important role in many physiological and developmental processes including stress response. Bacterial effects on plant hormonal system are important not only in promoting plant growth directly, but also in other aspects of bacterial action on plants. Thus hormone-mediated stimulation of root growth can improve mineral nutrition and water relations [17] Furthermore, hormone-mediated stimulation of plant antioxidant systems, helps to protect plants against oxidative stress, which accompanies action of most of detrimental environmental factors on plants [18]”.

  1. “Make a consistency throughout the manuscript by using single notation for each word for e.g. stress-free condition/without PEG/in the absence of PEG was used at different places "119, 192; table 1". Same for PEG TREATED/mild osmotic stress "line 170" as well as IAA content/auxin content

– In accordance with this remark we tried to use mainly “with or without PEG” and “PEG-treated” in the tables, figure legends and Result section. Numerous alterations may be tracked.

  1. “On what basis have you shortlisted the time points for the in-vitro extraction of bacterial hormones, Table 2?”

– In accordance with this remark we added to the description of results presented in Table 2 (lines 160-161) that : “For A. brasilense Sp245 and O. cytisi IPA7.2 cultures, 36 and 102 hours corresponded to the early and late stationary phases of growth”

  1. “Its not clear, Reconstruct the line " Furthermore, unlike A. brasilense........." Line 136-138.

 – in accordance with this remark we modified the sentence in hope that this variant may better: “Furthermore, in the case of growing plants in solution with PEG, inoculation with A. brasilense Sp245 did not influence root mass, while O. cytisi IPA7.2 increased root biomass accumulation compared to the plants non-inoculated with bacteria.”

  1. “Authors need to shed more light in discussion section on role of specific phytohormone as far as possible in the light of already reported information/mechanism of each stress responsive phytohormone in various plants.”

– In accordance, we added several sentence concerning hormones we have studied (auxins ABA and cytokinins):

  • Lines 259-263. “Our data are in accordance with those published previously showing that auxin producing licheniformis reduced the extent of growth inhibition by drought stress in pepper [41]. The increase in drought resistance by auxins have been also shown in transgenic potato plants overexpressing Arabidopsis YUCCA gene responsible for auxin synthesis [42].”
  • Lines 272-274. “ABA accumulates to high levels during drought stress [17, and references therein]. Production of this hormone by Azospirillum lipoferum increased ABA concentration in inoculated maize (Z. mays) seedlings [44] resulting in stomatal closure. However, stomatal closure inhibits photosynthesis leading to inhibition of plant growth. Consequently, accumulation of ABA in the leaves of osmotically stressed plants observed in the absence of bacteria (Fig. 2) could be the cause of inhibition of leaf growth, while suppression of the PEG-induced increase in the leaf abscisic acid by bacterial inoculation may contribute to their faster growth. Our data are in accordance with those showing that Bacillus subtilis maintained photosynthesis and growth of Arabidopsis plants by reducing ABA concentration in planta [45].”
  • Lines 283-286. “Known ability of cytokinins to maintain plant growth either under normal or stress conditions [46, and references therein] is important for controlling plant productivity.”

Round 2

Reviewer 1 Report

Dear Editor,
the authors accepted my suggestions and made the corrections. I therefore consider the manuscript acceptable for publication.